# DAG-aware GAT for Causal Effect Estimation

**Manqing Liu**  *manqingliu@g.havard.edu*
*Department of Epidemiology*
*Harvard University*

**David R. Bellamy**  *bellamyrd@gmail.com*

**Andrew L. Beam**  *andrew_beam@hms.harvard.edu*
*Department of Epidemiology*
*Harvard University*

**Reviewed on OpenReview:** *https://openreview.net/forum?id=XXXX*

## Abstract

Causal inference from observational data is a cornerstone of decision-making in healthcare, economics, and the social sciences. While deep learning has significantly advanced effect estimation, standard architectures often fail to respect the structural constraints inherent in causal systems, leading to biased results in complex scenarios like proximal inference. In this paper, we introduce the **DAG-aware Graph Attention Network (GAT)**, a novel neural framework that bridges structural causal modeling with graph representation learning. Unlike traditional Transformers or unconstrained GNNs, our model embeds the causal Directed Acyclic Graph (DAG) as a hard structural inductive bias directly into the attention mechanism. This ensures that information flow strictly adheres to valid causal pathways while preserving the semantic integrity of heterogeneous variables by omitting distortive normalization layers. Extensive experiments on several benchmark datasets show that the DAG-aware GAT consistently outperforms classical non-parametric baselines, modular MLPs, and causally-agnostic graph architectures. By prioritizing causal integrity over generic predictive heuristics, our approach provides a robust and interpretable foundation for reasoning from complex observational data. Our implementation is available at `https://github.com/ManqingLiu/DAG-aware-GAT`.

## 1 Introduction

The estimation of Average Treatment Effect (ATE) and Conditional Average Treatment Effect (CATE) is fundamental to data-driven decision-making in fields ranging from personalized medicine to economic policy Hernán & Robins (2024); Glass et al. (2013); Angrist & Pischke (2008). A primary challenge in these domains is the sensitivity of traditional estimators—such as Inverse Probability of Treatment Weighting (IPTW) and Doubly-Robust methods—to model misspecification Kang & Schafer (2007); Funk et al. (2011). This vulnerability is compounded in real-world scenarios by high-dimensional data and complex, often non-linear, causal dependencies Chernozhukov et al. (2018); Wager & Athey (2018).

To address these complexities, machine learning (ML) and deep learning (DL) methods have been increasingly integrated into causal frameworks. Early adaptations included causal forests Athey & Imbens (2016); Wager & Athey (2018) and double machine learning Chernozhukov et al. (2018), followed by representation learning techniques to balance covariates Shalit et al. (2017); Louizos et al. (2017). More recently, Graph Neural Networks (GNNs) and Transformer architectures have shown promise in capturing complex covariate interactions and long-range dependencies Ma & Tresp (2020); Guo et al. (2021); Melnychuk et al. (2022); Zhang et al. (2023).

Despite these advancements, a significant **causal-representational gap** persists. Standard DL architectures are typically "causally agnostic," treating inputs as flat vectors or fully-connected graphs where any variable can influence any other. Without structural guidance, high-capacity models often overfit to spurious correlations or allow "information leakage" across causal boundaries that should remain independent according to the underlying Directed Acyclic Graph (DAG). This lack of constraint means that even if a model is provided with the correct subset of variables, its internal representations may still contradict the system's causal topology.

This gap is particularly critical in **proximal causal inference** (Miao et al., 2018), which utilizes outcome proxies ($W$) and treatment proxies ($Z$) to handle unmeasured confounding. Kompa et al. (2022) introduced the Neural Maximum Moment Restriction (NMMR) framework, providing a robust statistical objective for learning the bridge function $h$ via neural networks. However, a significant limitation remains: NMMR utilizes a standard Multi-Layer Perceptron (MLP) as its estimator. As a causally-agnostic architecture, the MLP cannot distinguish between the asymmetric causal roles of the proxies or enforce the path constraints defined by the DAG.

To bridge this gap, we introduce a **DAG-aware Graph Attention Network (GAT)**. Our primary contribution is the replacement of causally-agnostic estimators with a structured architecture that embeds the causal topology directly into the model's structural inductive bias. By masking attention scores based on the DAG's adjacency matrix, we ensure that the representation of each variable—including the bridge function components—is strictly a function of its causal ancestors. This architectural constraint prevents the model from learning relationships that contradict the DAG, ensuring the resulting embeddings are causally-consistent.

Our framework enables the unified estimation of the propensity score $P(A|\mathbf{X})$, the outcome regression $P(Y|A, \mathbf{X})$, and the proximal bridge function $h(A, W, \mathbf{X})$ within a single, structurally-grounded model. Our contributions include: (1) a DAG-aware GAT architecture that resolves the causal-representational gap by encoding causal relationships directly into the attention mechanism; (2) unified estimation of multiple causal quantities; (3) native support for proximal inference via specialized proxy modeling; and (4) empirical validation demonstrating superior robustness to structural misspecification compared to existing methods.

## 2 Preliminaries

### 2.1 ATE and CATE

Consider treatment $A$ and its effect on outcome $Y$. Let $\mathbf{X}$ denote a vector of *observed* confounders. We define $Y^a$ as the counterfactual outcome for each individual had they received ($a = 1$) or not received ($a = 0$) the treatment. The Average Treatment Effect (ATE), denoted as $\tau$, is then defined as $\tau = \mathbb{E}[Y^1 - Y^0]$.

While the ATE provides an overall measure of the treatment effect across the entire population, in many cases, it's important to understand how the treatment effect varies across different subgroups or individuals. The CATE, denoted as $\tau(x)$, measures the average treatment effect for a subpopulation with a specific set of covariates $X = x$: $\tau(x) = \mathbb{E}[Y^1 - Y^0|X = x]$.

### 2.2 Confounding Control Methods assuming Unconfoundedness

In causal inference, several methods have been developed to control for *observed* confounding and estimate treatment effects. Our paper focuses primarily on three methods: Standardization (G-formula), Inverse Probability of Treatment Weighting (IPTW) and Augmented Inverse Probability Weighting (AIPW), a form of Doubly Robust estimator (Hernán & Robins, 2024).

1. **Standardization (G-formula)**: Standardization, also known as the G-formula, estimates the ATE by modeling the outcome as a function of treatment and confounders. It then averages over the confounder distribution to estimate the population-level effect. The ATE is estimated as:

$$\tau_G = \mathbb{E}_X[\mathbb{E}[Y|A = 1, X] - \mathbb{E}[Y|A = 0, X]] \tag{1}$$

where $\mu(a, X) = \mathbb{E}[Y|A = a, X]$ is the conditional expectation of the outcome given treatment $a$ and confounders $X$. This method is effective when the outcome model is correctly specified.

2. **Inverse Probability of Treatment Weighting (IPTW)**: IPTW uses the propensity score to create a pseudo-population in which the treatment assignment is independent of the measured confounders. The ATE is estimated as:

$$\tau_{IPTW} = \mathbb{E}\Big[\frac{AY}{\pi(X)} - \frac{(1-A)Y}{1-\pi(X)}\Big] \tag{2}$$

where $\pi(X) = P(A = 1|X)$ is the propensity score. This method is effective when the propensity score model is correctly specified.

3. **Augmented Inverse Probability Weighting (AIPW)**: AIPW combines IPTW with an outcome regression model, providing robustness against misspecification of either the propensity score model or the outcome model. The ATE is estimated as:

$$\begin{aligned}
\tau_{AIPW} = \mathbb{E}\Big[&\big(\mu(1, X) + \frac{A}{\pi(X)}(Y - \mu(1, X))\big) \\
&- \big(\mu(0, X) + \frac{1-A}{1-\pi(X)}(Y - \mu(0, X))\big)\Big]
\end{aligned} \tag{3}$$

where $\mu(a, X) = \mathbb{E}[Y|A = a, X]$ is the outcome regression function and $\pi(X) = P(A = 1|X)$ is the propensity score.

## 2.3 Proximal Inference

Proximal causal inference (Tchetgen et al., 2024) addresses unobserved confounding $U$ by utilizing a *tuple* of proxies $(W, Z)$ rather than a single proxy. The roles of these proxies are fundamentally asymmetric: $W$ is an **outcome proxy** (a "negative control outcome" that is a noisy measure of $U$), while $Z$ is a **treatment proxy** (a "negative control exposure" used to anchor the variation in $U$). This pair is necessary because a single proxy cannot simultaneously account for $U$'s effect on both the treatment and the outcome; $Z$ effectively acts as an instrumental variable to identify the "bridge" between $W$ and the latent $U$.

**Assumption 1** (Independence). $Y \perp\!\!\!\perp Z \mid A, U, X$ and $W \perp\!\!\!\perp (A, Z) \mid U, X$. *Note: $Z$ affects $Y$ only through $(A, U)$, whereas $W$ is independent of the treatment mechanism once $U$ is known.*

**Assumption 2 & 3** (Completeness). For all $f \in L^2$ and $a, x$:

- $\mathbb{E}[f(U)|a, x, z] = 0 \iff f(U) = 0$ almost surely.

- $\mathbb{E}[f(Z)|a, w, x] = 0 \iff f(Z) = 0$ almost surely.

Here, $L^2$ denotes the Hilbert space of square-integrable functions (i.e., $\mathbb{E}[f^2] < \infty$). These conditions ensure that the proxies are "rich" enough to capture all relevant variation in the latent $U$. Specifically, they ensure the conditional expectation operator is injective, allowing us to uniquely invert the relationship and identify the bridge function $h$ via:

$$\mathbb{E}[Y|A = a, X = x, Z = z] = \int_{\mathcal{W}} h(a, w, x)p(w|a, x, z)dw \tag{4}$$

The potential outcome is identified by: $\mathbb{E}[Y^a] = \mathbb{E}_{W,X}[h(a, W, X)]$, where the ATE is the empirical mean: $\hat{\mathbb{E}}[Y^a] = \frac{1}{M}\sum_{i=1}^{M} \hat{h}(a, w_i, x_i)$.

## 3   Methodology

We propose a novel DAG-aware GAT model for causal effect estimation that explicitly incorporates causal structure into the attention mechanism. Our approach is flexible and can accommodate various causal scenarios, including those with or without unmeasured confounding.

Given a dataset of $N$ observations, we define a set of possible input nodes. These include: $A$, the treatment variable; $\mathbf{X}$, the observed confounding variables; $U$, representing unmeasured confounding variables; $Y$, the outcome variable; $Z$, the proxy variable for treatment; and $W$, the proxy variable for outcome. The specific combination of input nodes used depends on the causal structure being modeled. The output nodes of our model vary based on the estimation method employed:

- For standardization or proximal inference: $\hat{Y}$ (estimated outcome), which corresponds to $\hat{\mu}(a, X)$ for standardization or $\hat{h}(a, W, X)$ for proximal inference.

- For Inverse Probability of Treatment Weighting (IPTW): $\hat{A}$ (estimated treatment probability), which corresponds to $\hat{\pi}(X)$ (estimated propensity score)

- For Augmented Inverse Probability Weighting (AIPW): Both $\hat{A}$ and $\hat{Y}$, where $\hat{A}$ corresponds to $\hat{\pi}(X)$ and $\hat{Y}$ corresponds to $\hat{\mu}(a, X)$

This flexible framework allows our model to adapt to different causal inference scenarios and estimation techniques while maintaining its core structure.

After estimating these quantities, we can plug them into the corresponding formulas from Section 2 to estimate the Average Treatment Effect (ATE) or Conditional Average Treatment Effect (CATE). Specifically, we replace the true (but unknown) functions $\pi(X)$ and $\mu(a, X)$ with their estimates $\hat{\pi}(X)$ and $\hat{\mu}(a, X)$ obtained from our model:

- **For standardization**: We estimate $\hat{\mu}(a, X) = \mathbb{E}[Y|A = a, X]$ using our model, then compute:

$$\hat{\tau}_G = \frac{1}{N} \sum_{i=1}^{N} [\hat{\mu}(1, X_i) - \hat{\mu}(0, X_i)] \tag{5}$$

- **For IPTW**: We estimate $\hat{\pi}(X) = P(A = 1|X)$ using our model, then compute:

$$\hat{\tau}_{IPTW} = \frac{1}{N} \sum_{i=1}^{N} \left[ \frac{A_i Y_i}{\hat{\pi}(X_i)} - \frac{(1 - A_i)Y_i}{1 - \hat{\pi}(X_i)} \right] \tag{6}$$

- **For AIPW**: We estimate both $\hat{\pi}(X)$ and $\hat{\mu}(a, X)$ using our model. These estimates replace the true functions in Equation 3:

$$\begin{aligned} \hat{\tau}_{AIPW} = \frac{1}{N} \sum_{i=1}^{N} \Big[ &\left( \hat{\mu}(1, X_i) + \frac{A_i}{\hat{\pi}(X_i)}(Y_i - \hat{\mu}(1, X_i)) \right) \\ &- \left( \hat{\mu}(0, X_i) + \frac{1 - A_i}{1 - \hat{\pi}(X_i)}(Y_i - \hat{\mu}(0, X_i)) \right) \Big] \end{aligned} \tag{7}$$

where $\hat{\mu}(a, X)$ is our model's estimate of $\mathbb{E}[Y|A = a, X]$ and $\hat{\pi}(X)$ is our model's estimate of $P(A = 1|X)$.

- **For proximal inference**: We estimate the bridge function $\hat{h}(a, W, X)$ using our model, then compute:

$$\hat{\tau}_{proximal} = \frac{1}{N} \sum_{i=1}^{N} [\hat{h}(1, W_i, X_i) - \hat{h}(0, W_i, X_i)] \tag{8}$$

For CATE estimation, we can condition on specific values of $X$ in these equations. This approach allows us to estimate both population-level and subgroup-level causal effects using our DAG-aware Transformer model, with the key advantage that all required functions ($\pi(X)$, $\mu(a, X)$, or $h(a, W, X)$) are estimated within a unified framework that respects the causal structure of the problem.

## 3.1 DAG-aware GAT Architecture

We model the causal system as a Graph Attention Network (GAT) where each node represents a distinct causal variable (e.g., covariates, treatment, or proxies). Unlike standard Transformers that process homogeneous tokens (e.g., word embeddings), our architecture handles heterogeneous variables with varying semantic scales. To preserve these distinct distributions, we utilize a GAT Encoder that implements DAG-constrained multi-head attention without the use of Layer Normalization. While Layer Normalization is standard in sequence-to-sequence Transformers to stabilize homogeneous embeddings, applying it across heterogeneous causal variables (such as binary treatment indicators and continuous income scales) would inappropriately force them into a shared distribution, thereby inducing empirical bias in causal effect estimation. Our GAT-based approach avoids this by maintaining the independent feature scales while leveraging the expressive power of the attention mechanism.

Figure 1 illustrates our model architecture with a simple example DAG containing treatment ($A$), confounders ($X$), and outcome ($Y$) nodes. While this figure shows a basic scenario for clarity, our architecture is flexible and can accommodate more complex causal structures, including proximal inference settings with additional proxy nodes ($W$ and $Z$) (See Figure 3 for an example), time-varying treatments, or high-dimensional confounder spaces.

We encode the causal DAG into an adjacency matrix $\mathbf{M}^{adj} \in \{0, 1\}^{D \times D}$, where $D$ is the number of nodes in the graph. Each element $M_{ij}^{adj} = 1$ indicates a directed edge from node $i$ to node $j$, representing a direct causal relationship. Importantly, we enforce $M_{ii}^{adj} = 0$ for all $i$, ensuring no self-loops exist in the causal graph, as nodes cannot cause themselves.

To incorporate this causal structure into the attention mechanism, we transform the adjacency matrix into an attention mask $\mathbf{M}$:

$$M_{ij} = \begin{cases} 0 & \text{if } M_{ji}^{adj} = 1 \\ 1 & \text{otherwise} \end{cases} \tag{9}$$

Our key innovation lies in incorporating the causal structure directly into the multi-head attention computation. For each attention head, we first compute the standard attention scores:

$$\mathbf{A} = \frac{\mathbf{Q}\mathbf{K}^T}{\sqrt{E}} \tag{10}$$

where $\mathbf{Q}, \mathbf{K} \in \mathbb{R}^{N \times D \times E}$ are the query and key matrices respectively, $N$ is the batch size, and $E$ is the embedding dimension.

We then apply the DAG-based mask to these attention scores:

$$\mathbf{A}^{mask} = \mathbf{A} + \mathbf{M} \cdot (-\infty) \tag{11}$$

This masking operation effectively sets attention scores to negative infinity for node pairs that are

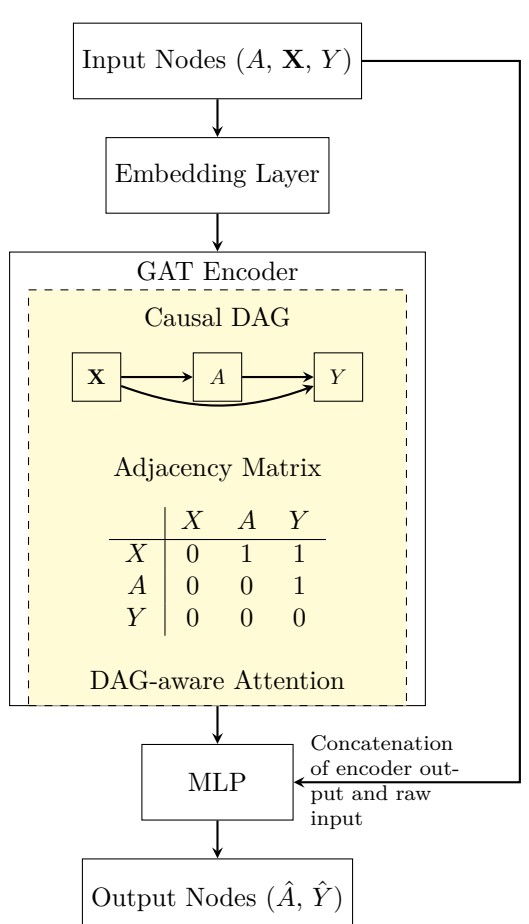

Figure 1: Architecture of the DAG-aware GAT model. The GAT Encoder utilizes a DAG-constrained attention mechanism (indicated by dashed lines) to capture the causal dependencies defined by the graph's

not causally connected according to the DAG. After applying softmax, these masked positions will have zero attention weight, preventing information flow between causally unrelated nodes.

The final attention output is computed as:

$$\text{Attention}(\mathbf{Q}, \mathbf{K}, \mathbf{V}) = \text{softmax}(\mathbf{A}^{mask})\mathbf{V} \qquad (12)$$

where $\mathbf{V} \in \mathbb{R}^{N \times D \times E}$ is the value matrix.

Figure 2 illustrates the visualization of the adjacency matrix for a simple scenario involving treatment $A$, two confounders $X_1$ and $X_2$, and the outcome $Y$. Additionally, it features the corresponding masking matrix and the attention weights, providing insights into the attention mechanism used in the model.

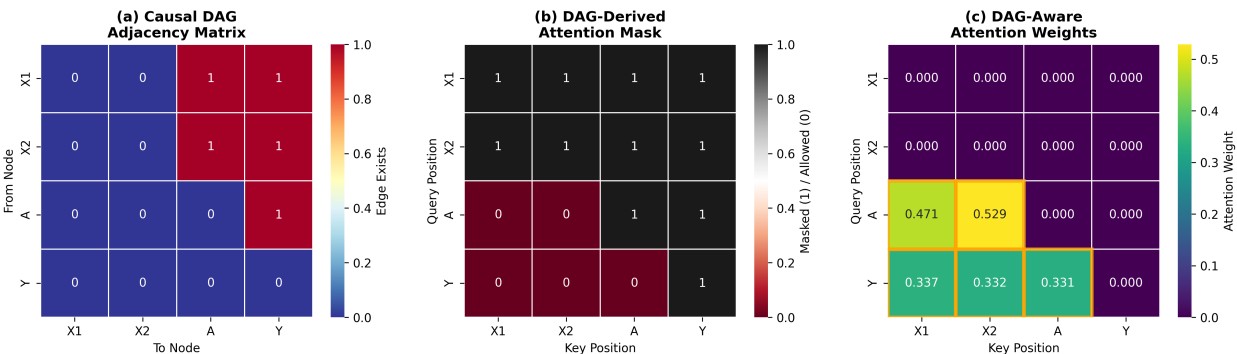

Figure 2: Visualization of key components in the DAG-aware GAT model. (a) Causal DAG Adjacency Matrix: This matrix represents the directed edges between nodes, indicating the causal relationships among features. A value of 1 denotes the existence of an edge, while 0 signifies no direct relationship. (b) DAG-Derived Attention Mask: The attention mask, derived from the adjacency matrix, specifies which nodes can attend to one another during the attention computation. Here, a value of 0 indicates that attention is allowed, while 1 indicates masked positions, preventing information flow between non-causal nodes. (c) DAG-Aware Attention Weights: These weights illustrate the strength of attention assigned to each node pair in the attention mechanism, reflecting how the model weighs the importance of different nodes when updating information. The color scale indicates varying levels of attention, with brighter regions representing higher weights.

After processing inputs through multiple layers of DAG-aware attention, we combine the GAT encoder's output $\mathbf{H}$ with the original features $\mathbf{X}$ through a hybrid integration mechanism:

$$\mathbf{Z} = \alpha \cdot \mathbf{H} + \mathbf{X} \qquad (13)$$

where $\alpha$ is a learnable weight parameter. This design, inspired by residual connections He et al. (2016), addresses an empirical phenomenon we observed: relying solely on the GAT's representations can lose critical confounding information, causing the model to produce identical outcome predictions regardless of treatment assignment. By preserving both the learned representations and raw features, the hybrid approach ensures that essential confounding information remains accessible for downstream prediction.

The hybrid representation $\mathbf{Z}$ is then processed by a Multi-Layer Perceptron (MLP) to produce the final predictions:

$$\hat{A}/\hat{Y} = \text{MLP}(\mathbf{Z}) \qquad (14)$$

The specific output depends on the causal inference task: $\hat{A}$ for propensity score estimation in IPTW, $\hat{Y}$ for outcome regression in standardization, or both for AIPW. In proximal inference settings with unmeasured confounding, the model estimates the bridge function $\hat{h}(a, W, X)$ instead.

**DAG Specification and Discovery.** Our framework assumes that a causal DAG is provided as a prior, a common requirement in structural causal modeling where graph topology is derived from domain expertise or established literature. However, in scenarios where the causal structure is unknown, a DAG can be obtained using causal discovery algorithms such as PC (Spirtes et al., 2000), GES (Chickering, 2002), or gradient-based methods like NOTEARS (Zheng et al., 2018).

When learning a DAG from data, we caution against using the same dataset for both structure discovery and effect estimation to avoid post-selection bias and overfitting. We recommend a sample-splitting approach: a discovery set is used to identify the DAG, and a separate estimation set is used to train our DAG-aware GAT. This ensures that the structural constraints imposed on the attention mechanism are not inadvertently capturing noise specific to the estimation sample. In our experiments, we utilize established benchmark DAGs to focus on the estimation performance, but the model is fully compatible with any valid DAG identified through external discovery processes.

**DAG Integration vs. Feature Selection.** A natural question arises as to why standard architectures cannot simply leverage DAG knowledge through external pre-processing, such as feature selection based on adjustment sets. While traditional methods (e.g., MLPs or standard Transformers) can be restricted to specific variable subsets, they treat the input as a "flat" vector or a fully-connected graph, respectively. Consequently, they lack the structural inductive bias to respect the specific directional dependencies and path constraints defined by the DAG. In contrast, our DAG-aware attention mechanism embeds the causal topology directly into the model's internal information flow. By masking attention scores based on the adjacency matrix, we ensure that the representation of each node is strictly a function of its causal ancestors. This architectural constraint prevents the model from "learning" spurious correlations that contradict the DAG, a guarantee that cannot be enforced in standard architectures even if they are provided with the correct subset of variables.

### 3.2 Model Training and Objective Function

Our model employs different loss functions depending on the causal inference method used. We present the loss functions for standardization (G-formula), Inverse Probability of Treatment Weighting (IPTW), Augmented Inverse Probability Weighting (AIPW), and proximal inference. Here we assume $Y$ (outcome) is continuous, and $A$ (treatment) is a binary variable.

For standardization, we use Mean Squared Error (MSE) loss for maximum likelihood estimation:

$$\mathcal{L}_G = \text{MSE}(\hat{Y}, Y) = \frac{1}{n} \sum_{i=1}^{n} (\hat{Y}_i - Y_i)^2 \tag{15}$$

where $\hat{Y}$ are the model outputs and $Y$ are the true labels.

For IPTW, we use Binary Cross Entropy (BCE) loss for treatment/propensity score estimation:

$$\mathcal{L}_{\text{IPTW}} = \text{BCE}(\hat{A}, A) =$$
$$-\frac{1}{n} \sum_{i=1}^{n} [A_i \log(\hat{A}_i) + (1 - A_i) \log(1 - \hat{A}_i)] \tag{16}$$

where $\hat{A}$ are the model outputs (estimated propensity scores) and $A$ are the true treatment assignments.

For AIPW, we combine MSE loss for outcome prediction and BCE loss for treatment assignment:

$$\mathcal{L}_{\text{AIPW}} = \frac{1}{2}(\text{MSE}(\hat{Y}, Y) + \text{BCE}(\hat{A}, A)) \tag{17}$$

### 3.3 Neural Maximum Moment Restriction (NMMR)

To estimate the bridge function $h$ in the proximal framework, we utilize the **Neural Maximum Moment Restriction (NMMR)** approach (Kompa et al., 2022). This method identifies $h$ by minimizing the conditional moment restriction $\mathbb{E}[Y - h(A, W, X)|A, X, Z] = 0$. In NMMR, this restriction is mapped into a Reproducing Kernel Hilbert Space (RKHS) using a **kernel function** $k$, which measures the similarity between instances of the conditioning variables $(A, Z, X)$.

We define $k_{ij} = k((a_i, z_i, x_i), (a_j, z_j, x_j))$—typically an RBF kernel—and let $\mathbf{K}$ be the $n \times n$ kernel matrix (Gram matrix) with entries $k_{ij}$. Based on the theory of $U$ and $V$ statistics, we introduce two variants for the empirical risk $\hat{R}_{k,n}$ given data $\mathcal{D} = \{(a_i, w_i, x_i, y_i, z_i)\}_{i=1}^n$:

1. **NMMR-U**: Utilizes an unbiased $U$-statistic by setting the diagonal of $\mathbf{K}$ to zero ($k_{ii} = 0$ for all $i$), preventing the model from overemphasizing self-correlations:

$$\hat{R}_{k,U,n}(h) = \frac{1}{n(n-1)} \sum_{i,j=1, i \neq j}^n (y_i - h_i)(y_j - h_j)k_{ij} \tag{18}$$

2. **NMMR-V**: Utilizes a $V$-statistic by including the main diagonal of $\mathbf{K}$. While biased in finite samples, it is often more numerically stable during training:

$$\hat{R}_{k,V,n}(h) = \frac{1}{n^2} \sum_{i,j=1}^n (y_i - h_i)(y_j - h_j)k_{ij} \tag{19}$$

where $h_i = h(a_i, w_i, x_i)$. Here, $h$ is restricted to the space of square-integrable functions $L^2$ (i.e., $\mathbb{E}[f^2] < \infty$). This ensures the completeness assumptions (Assumptions 2 and 3) hold, allowing for the unique identification of $h$ via the proxies: $W$ serves as the outcome proxy while $Z$ serves as the treatment proxy required to "anchor" the integral equation.

To prevent overfitting, we add an $L_2$ penalty $\Lambda[h, \theta_h] = \|\theta_h\|_2^2$ on the network parameters $\theta_h$. The final loss function, represented in matrix form for efficient computation, is:

$$\mathcal{L}_{\text{proximal}} = (\mathbf{y} - \mathbf{h})^T \mathbf{K}(\mathbf{y} - \mathbf{h}) + \lambda \Lambda[h, \theta_h] \tag{20}$$

where $(\mathbf{y} - \mathbf{h})$ is the vector of residuals. Depending on the variant chosen, the diagonal of $\mathbf{K}$ is either preserved (NMMR-V) or masked (NMMR-U).

## 4 Experiments

We evaluate our proposed DAG-aware GAT across four diverse datasets, demonstrating its flexibility in handling different causal structures and estimation tasks. Our experiments are designed to showcase the key advantage of our approach: the ability to flexibly encode causal DAGs and estimate the conditional probability distributions required for various causal inference methods within a unified framework.

**Experimental Setup**. We employ four datasets that span different causal inference scenarios. The LaLonde CPS and PSID datasets are used for Average Treatment Effect (ATE) estimation in standard unconfoundedness settings, providing a benchmark for evaluating our method's performance on classic causal inference tasks. The ACIC dataset is employed for Conditional Average Treatment Effect (CATE) estimation, allowing us to assess our model's ability to capture heterogeneous treatment effects across different subpopulations. Additionally, we use the Demand dataset described in Kompa et al. (2022) for proximal inference scenarios with unmeasured confounding, demonstrating our model's ability to handle more complex causal structures

involving proxy variables. The causal assumptions and DAG structures for each experiment are detailed in Appendix A.1.

**Evaluation Metrics**. We employ normalized root mean square error (NRMSE) as our primary evaluation metric across all experiments. For the LaLonde and ACIC datasets, we compute NRMSE between our estimated ATE or CATE and the true values. In the case of the Demand dataset with proximal inference, performance is evaluated using NRMSE computed across 10 equally-spaced price points between 10 and 30, comparing our estimated potential outcomes $\hat{\mathbb{E}}[Y^a]$ against Monte Carlo simulations of the true $\mathbb{E}[Y^a]$. This comprehensive evaluation allows us to assess our model's performance across different causal inference scenarios and estimation targets.

**Baseline Models and Ablation Studies.** To evaluate the efficacy of our **DAG-aware GAT** model, we compare its performance against several established methods in the causal inference literature. We utilize Generalized Random Forests (GRF) (Wager & Athey, 2018) as a robust non-parametric baseline for estimating heterogeneous treatment effects. For deep learning benchmarks, we include: (i) a standard Multilayer Perceptron (MLP) to establish a feature-only baseline; (ii) a standard Graph Neural Network (GNN) that utilizes the DAG structure but lacks causal attention constraints; and (iii) a DAG-constrained Transformer, which incorporates layer normalization to specifically assess the empirical bias that normalization layers may introduce into causal effect estimation.

Furthermore, we conduct ablation studies to evaluate the model's sensitivity to structural misspecification through two distinct perturbations: (i) reversing the fundamental causal direction between treatment and outcome ($A \to Y$ transformed to $Y \to A$) (Figure 4), and (ii) removing the edge connecting the outcome proxy to the outcome ($W \to Y$), thereby breaking a critical identification condition (Figure 5). These scenarios allow us to quantify the structural robustness of the graph attention mechanism and observe how the enforcement of an incorrect causal topology affects the model's ability to recover valid effect estimates when core identification assumptions are violated.

While Transformer-based causal models such as CETransformer (Guo et al., 2021) and Causal Transformer (Melnychuk et al., 2022) have been proposed, they are tailored for specific tasks—namely balanced representation learning and longitudinal counterfactual estimation—and are not directly applicable to the diverse experimental settings explored here, which include both standard and proximal inference scenarios. Our work demonstrates that by reframing the problem as a graph-attention task, the GAT architecture can flexibly encode the explicit DAG structure to handle multiple causal inference frameworks within a single, unified model. Implementation details and hyperparameter configurations are provided in Appendix A.2.

## 5   Results

The empirical evaluation of the DAG-aware GAT across the LaLonde, ACIC, and Demand datasets demonstrates not only state-of-the-art performance but also provides critical insights into the intersection of deep learning architectures and causal topology.

**Architectural Alignment: GAT vs. Transformer** A primary result of our study is the significant performance gap between the DAG-aware GAT and the Transformer baseline, particularly in weighting-based estimators like IPW and AIPW. For instance, in the *LaLonde-CPS* dataset (Table 1), the GAT achieves an IPW NRMSE of 0.614, while the Transformer baseline, despite utilizing the same DAG-constrained attention, exhibits a degradation to 6.719. We interpret this failure as a consequence of **feature heterogeneity**. Standard Transformers are designed for homogeneous tokens (e.g., word embeddings) where Layer Normalization (LN) stabilizes the training by standardizing the semantic space. However, causal data is inherently heterogeneous; nodes represent distinct physical quantities (e.g., binary treatment indicators vs. continuous income scales). Applying LN forces these varying distributions into a shared mean and variance, effectively "smearing" the causal signals. By using the architecture of GAT without LN, we preserve the individual variance of each node. This preservation is essential for the high-fidelity density estimation required for propensity scores and causal bridge functions.

**Sensitivity to Structural Misspecification**   We evaluate the robustness of our framework by intentionally perturbing the causal prior on the *Demand* dataset. Following the proximal inference framework established by Kompa et al. (2022), our MLP baselines respect the underlying causal structure by separately estimating the required bridge functions. However, while the MLP utilizes the DAG to define its objective functions, the DAG-aware GAT embeds this topology directly into its internal message-passing mechanism.

Our results (Table 2) indicate that while a correct DAG provides a powerful inductive bias, a misspecified graph acts as a structural "ceiling" that prevents effective learning. At small sample sizes ($N = 1,000$), the performance gap between the correct DAG and misspecified variants is marginal, as the model likely relies on global feature correlations. However, as the sample size increases to $10,000$, the correctly specified GAT shows consistent convergence, whereas misspecified models—particularly those with reversed arrows—diverge sharply. In these scenarios, the GAT can perform worse than the modular MLP baseline. While the MLP remains robust by targeting valid identification objectives independently, the DAG-aware GAT is architecturally forced to aggregate features across invalid causal pathways. This confirms that while embedding structural constraints into the attention mechanism offers superior representational power, it necessitates a reliable causal prior, potentially sourced from expert knowledge or automated causal discovery pipelines.

## 6   Discussion and Conclusion

In this work, we introduced the **DAG-aware GAT**, a neural causal inference framework that bridges structural causal modeling with the representational power of Graph Attention Networks. Unlike transformer, our model is able to handle heterogeneous variables, such as continuous age, binary treatment indicators, and proxy variables, without the distortive effects of Layer Normalization. Our findings confirm that by omitting normalization layers, we preserve the distinct semantic scales of these variables, preventing the empirical bias that typically hinders the estimation of propensity scores and causal weights.

The core strength of our approach lies in using the causal DAG as a **hard structural inductive bias**. Unlike standard GNNs that learn graph topology for predictive tasks, our framework enforces a pre-defined or discovered DAG as a fixed constraint on the attention mechanism. This ensure that information flow strictly adheres to valid causal pathways and prevents "information leakage" between variables, such as proxies in proximal inference settings. While this framework relies on a provided DAG, it is highly complementary to emerging **causal foundation models** like SEA (Wu et al., 2025). While foundation models excel at identifying DAGs from observational data, our method serves as the essential estimation engine that respects those discovered constraints. By utilizing a sample-splitting strategy, where discovery models identify the DAG and our GAT performs the estimation, researchers can achieve a robust, end-to-end pipeline that avoids post-selection bias.

Our empirical results across the LaLonde, ACIC, and Demand datasets validate that incorporating explicit causal constraints into a GAT architecture provides superior performance compared to both classical non-parametric baselines and unconstrained deep learning models. By providing a principled framework that prioritizes causal integrity over generic predictive heuristics, our model demonstrates exceptional flexibility across diverse estimation strategies (G-formula, IPW, AIPW) and experimental settings. Future work may explore the integration of uncertainty quantification, the handling of time-varying treatments, and the joint optimization of structural discovery and effect estimation. As causal inference becomes increasingly vital for automated decision-making, architectures that embed structural guardrails directly into the attention mechanism will be essential for robust and interpretable reasoning from observational data.

Table 1: Performance comparison of causal inference estimators across datasets and model architectures.

| Dataset | Estimator | Model | NRMSE (mean) | NRMSE (SE) |
|---------|-----------|-------|--------------|------------|
| Lalonde-CPS | G-formula | GRF | 0.925 | 0.003 |
| | | MLP | 0.935 | 0.069 |
| | | GNN | 0.908 | 0.013 |
| | | Transformer (with Layernorm) | 0.906 | 0.012 |
| | | GAT (Ours) | **0.784** | **0.074** |
| | IPW | GRF | 6.342 | 1.227 |
| | | MLP | 6.419 | **0.471** |
| | | GNN | 5.839 | 0.496 |
| | | Transformer (with Layernorm) | 6.719 | 0.435 |
| | | GAT (Ours) | **0.614** | **0.100** |
| | AIPW | GRF | 1.596 | 0.294 |
| | | MLP | 1.362 | 0.216 |
| | | GNN | 0.883 | 0.615 |
| | | Transformer (with Layernorm) | 1.009 | 0.168 |
| | | GAT (Ours) | **0.351** | **0.130** |
| Lalonde-PSID | G-formula | GRF | 1.009 | 0.021 |
| | | MLP | 0.994 | 0.213 |
| | | GNN | 0.985 | **0.006** |
| | | Transformer (with Layernorm) | 0.985 | **0.006** |
| | | GAT (Ours) | **0.964** | 0.014 |
| | IPW | GRF | 9.408 | 1.108 |
| | | MLP | 1.850 | 0.818 |
| | | GNN | 2.238 | 0.878 |
| | | Transformer (with Layernorm) | 1.329 | 0.740 |
| | | GAT (Ours) | **0.257** | **0.093** |
| | AIPW | GRF | 2.517 | 0.242 |
| | | MLP | 2.038 | 0.482 |
| | | GNN | 1.757 | 0.491 |
| | | Transformer (with Layernorm) | 3.000 | 0.882 |
| | | GAT (Ours) | **0.922** | **0.264** |
| ACIC | G-formula | GRF | 0.346 | 0.044 |
| | | MLP | 0.558 | 0.062 |
| | | GNN | 0.555 | 0.092 |
| | | Transformer (with Layernorm) | 0.429 | 0.128 |
| | | GAT (Ours) | **0.344** | **0.054** |
| | IPW | GRF | 4.281 | 1.388 |
| | | MLP | 6.380 | 1.700 |
| | | GNN | 5.440 | 1.469 |
| | | Transformer (with Layernorm) | 4.858 | 1.397 |
| | | GAT (Ours) | **4.102** | **1.272** |
| | AIPW | GRF | 0.857 | 0.059 |
| | | MLP | 1.244 | 0.141 |
| | | GNN | 5.088 | 1.596 |
| | | Transformer (with Layernorm) | 3.853 | 1.252 |
| | | GAT (Ours) | **0.814** | **0.052** |

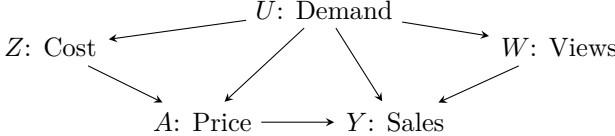

Figure 3: Causal DAG for the Demand experiment.

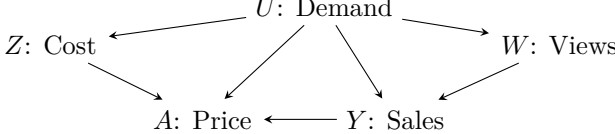

Figure 4: Mis-specified causal DAG by reversing the edge between treatment and outcome $(Y \to A)$

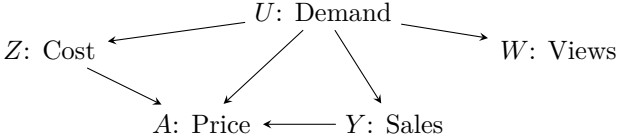

Figure 5: Mis-specified causal DAG by removing the arrow from $W \to Y$

Table 2: Performance comparison of U-statistics and V-statistics estimators on the Demand dataset across different sample sizes.

| Sample Size | Estimator | Model | NRMSE (mean) | NRMSE (SE) |
|---|---|---|---|---|
| 1,000 | U-statistics | NMMR (Kompa et al., 2022) | 0.127 | 0.008 |
| | | GAT (mis-specified DAG (reverse of arrow)) | 0.213 | 0.069 |
| | | GAT (mis-specified DAG (missing arrow)) | 1.149 | 0.024 |
| | | GAT (Ours) | **0.109** | **0.007** |
| | V-statistics | NMMR (Kompa et al., 2022) | 0.138 | 0.008 |
| | | GAT (mis-specified DAG) | 0.745 | 0.056 |
| | | GAT (mis-specified DAG (missing arrow)) | 0.807 | 0.108 |
| | | GAT (Ours) | **0.131** | **0.007** |
| 5,000 | U-statistics | NMMR (Kompa et al., 2022) | 0.114 | 0.006 |
| | | GAT (mis-specified DAG (reverse of arrow)) | 1.642 | 0.032 |
| | | GAT (mis-specified DAG (missing arrow)) | 1.641 | 0.029 |
| | | GAT (Ours) | **0.096** | **0.005** |
| | V-statistics | NMMR (Kompa et al., 2022) | 0.126 | 0.006 |
| | | GAT (mis-specified DAG (reverse of arrow)) | 0.700 | 0.077 |
| | | GAT (mis-specified DAG (missing arrow)) | 0.809 | 0.069 |
| | | GAT (Ours) | **0.108** | **0.006** |
| 10,000 | U-statistics | NMMR (Kompa et al., 2022) | 0.137 | 0.012 |
| | | GAT (mis-specified DAG (reverse of arrow)) | 1.523 | 0.057 |
| | | GAT (mis-specified DAG (missing arrow)) | 1.447 | 0.068 |
| | | GAT (Ours) | **0.102** | **0.004** |
| | V-statistics | NMMR (Kompa et al., 2022) | 0.132 | 0.007 |
| | | GAT (mis-specified DAG (reverse of arrow)) | 0.815 | 0.077 |
| | | GAT (mis-specified DAG (missing arrow)) | 0.903 | 0.056 |
| | | GAT (Ours) | **0.103** | **0.004** |

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

# A Appendix

## A.1 Causal Assumptions

To ensure valid causal inference, several key assumptions must hold. In this paper, we primarily focus on three fundamental assumptions:

1. **Positivity (or Overlap)**: For every $x \in \text{support}(X)$, and $\forall a \in \{0, 1\}$, $P(A = a|X = x) > 0$.

   This assumption ensures that there is a non-zero probability of receiving each treatment level for all possible values of the observed covariates. It is crucial for estimating treatment effects across the entire covariate space and prevents extrapolation to regions where we have no information about one of the treatment groups.

2. **Exchangeability (or Unconfoundedness)**: $Y^a \perp\!\!\!\perp A|X, \forall a \in \{0, 1\}$.

   This assumption implies that, conditional on the observed confounders $X$, the potential outcomes $Y^a$ are independent of the treatment assignment $A$. In other words, after controlling for $X$, there are no unmeasured confounders that affect both the treatment assignment and the outcome. This is also known as the "no unmeasured confounding" assumption.

3. **Consistency**: If $A = a$, then $Y^a = Y$.

   This assumption states that the potential outcome under a particular treatment level is the same as the observed outcome if the individual actually receives that treatment level. It ensures that the observed outcomes can be used to estimate the potential outcomes.

For the Lalonde and ACIC experiments, we assume that all three of these assumptions hold. For the proximal infererence experiment, we remove the strong assumption of no unmeasured confounding (Assumption 2).

## A.2 Model Selection and Hyperparameter Tuning

A central challenge in causal inference is that we cannot directly observe counterfactual potential outcomes $Y^a$ for the same individual across different treatment levels (Saito & Yasui, 2020). Consequently, traditional cross-validation using observed labels is insufficient for model selection. While our training objective (NMMR) relies on observed data through the proximal bridge identity, selecting the optimal hyperparameters requires a surrogate for the true causal effect.

Following Saito & Yasui (2020) and Mahajan et al. (2024), we employ a "plug-in" estimation strategy to approximate the missing ground truth. Our procedure is as follows:

1. **Surrogate Estimation:** We train a "quasi-oracle" plug-in estimator $\hat{\tau}$ on the validation set using Generalized Random Forests (GRF) (Wager & Athey, 2018). This provides a high-capacity, non-parametric baseline for the Average Treatment Effect (ATE) or Conditional Average Treatment Effect (CATE).

2. **Model Ranking:** We evaluate our candidate proximal estimators $\tilde{\tau} \in \mathcal{T}$ by calculating their deviation from the surrogate $\hat{\tau}$.

Formally, we select the optimal estimator $\tilde{\tau}^*$ by minimizing the Normalized Root Mean Squared Error (NRMSE):

$$\tilde{\tau}^* = \underset{\tilde{\tau} \in \mathcal{T}}{\arg\min} \ \text{NRMSE} \left( \hat{\tau}, \tilde{\tau} \right) \tag{21}$$

The NRMSE is defined as:

$$\text{NRMSE} = \sqrt{\frac{\frac{1}{n-1} \sum_{i=1}^{n} (\hat{\tau}(X_i) - \tilde{\tau}(X_i))^2}{\hat{V}(\hat{\tau}(X))}} \tag{22}$$

where $\{\tilde{\tau}(X_i)\}_{i=1}^n$ are the predictions generated by our model, and $\hat{V}(\hat{\tau}(X))$ is the empirical variance of the surrogate predictions, which serves as a normalization factor to ensure the metric is scale-invariant across different datasets.

We performed extensive hyperparameter tuning for each dataset and estimation method. The hyperparameter spaces explored are detailed below. For each configuration, we report the range of values considered during the tuning process.

Table 3: Hyperparameter tuning ranges for Lalonde-CPS dataset

| Parameter | G-formula | IPW | AIPW |
|---|---|---|---|
| Number of epochs | 80 | 20 | 20 |
| Batch size | 32 | 32 | 32 |
| Learning rate | 0.001 | 0.001 | 0.001 |
| L2 penalty | 3e-05, 3e-03 | 3e-05, 3e-03 | 3e-05, 3e-03 |
| Network width (MLP) | 80 | 80 | 80 |
| Input layer depth (MLP) | 2–4 | 1–2 | 2–6 |
| Number of layers (encoder) | 2–4 | 1–2 | 1–2 |
| Dropout rate | 0.0001 | 0.0001–0.001 | 0.0001 |
| Embedding dimension (encoder) | 40 | 40 | 40 |
| Feedforward dimension (encoder) | 80 | 80 | 80 |
| Number of heads (encoder) | 2 | 1–2 | 1–2 |
| Encoder weight (alpha) | 0.02 | 0.002–0.02 | 0.02 |

Table 4: Hyperparameter tuning ranges for Lalonde-PSID dataset

| Parameter | G-formula | IPW | AIPW |
|---|---|---|---|
| Number of epochs | 100 | 30 | 30 |
| Batch size | 32, 64 | 64 | 32, 64 |
| Learning rate | 0.001–0.01 | 0.001 | 0.001 |
| L2 penalty | 3e-05 | 3e-05 | 3e-05–3e-03 |
| Network width (MLP) | 80 | 10, 20 | 40 |
| Input layer depth (MLP) | 6–16 | 1 | 4–8 |
| Number of layers (encoder) | 1–2 | 1 | 1–2 |
| Dropout rate | 0.0001 | 0.0001 | 0.0001 |
| Embedding dimension (encoder) | 40 | 10–40 | 20–40 |
| Feedforward dimension (encoder) | 80 | 20–80 | 40–80 |
| Number of heads (encoder) | 1–2 | 1 | 2 |
| Encoder weight (alpha) | 0.002–0.02 | 0.02–0.2 | 0.02–0.2 |

For each dataset and estimation method, we performed a grid search over these hyperparameter spaces. The final model for each configuration was selected based on the best performance on a held-out validation set.

Table 5: Hyperparameter tuning ranges for ACIC dataset

| Parameter | G-formula | IPW | AIPW |
|---|---|---|---|
| Number of epochs | 500 | 30 | 500 |
| Batch size | 64 | 64, 128, 256 | 64, 128, 256 |
| Learning rate | 1e-03 | 1e-03 | 1e-04 |
| L2 penalty | 3e-08 | 3e-05 | 3e-08, 3e-04 |
| Network width (MLP) | 40 | 40, 60, 80 | 40, 60, 80, 120 |
| Input layer depth (MLP) | 8–16 | 4–16 | 2–16 |
| Number of layers (encoder) | 16 | 2–4 | 2–8 |
| Dropout rate | 0.0001 | 0.0001 | 0.0001–0.003 |
| Embedding dimension (encoder) | 256 | 40, 60, 80 | 256 |
| Feedforward dimension (encoder) | 1024 | 80, 320 | 512, 1024 |
| Number of heads (encoder) | 4 | 2 | 2–4 |
| Encoder weight (alpha) | 0.02–0.2 | 0.002–0.2 | 0.1–2 |

Table 6: Hyperparameter tuning ranges for Demand dataset (Proximal Inference)

| Parameter | Range |
|---|---|
| Number of epochs | 1000 |
| Batch size | 32, 64 |
| Learning rate | 0.001 |
| L2 penalty | 3e-06 |
| Network width (MLP) | 160 |
| Input layer depth (MLP) | 8, 16 |
| Number of layers (encoder) | 1, 2 |
| Dropout rate | 0 |
| Embedding dimension (encoder) | 40 |
| Feedforward dimension (encoder) | 40, 80 |
| Number of heads (encoder) | 1, 2 |
| Encoder weight (alpha) | 0.001–0.025 |

