# OpenReview forum: "DAG-aware GAT for Causal Effect Estimation"
_TMLR — Accepted by TMLR_

### Review · Reviewer_1zXp · 2025-12-15

**Summary Of Contributions:**

The paper proposes a "DAG-aware Transformer", a unified framework for estimating causal effects (ATE and CATE) via propensity scores and regressions. The core innovation is a masked self-attention mechanism which limits attention to only a priori known direct causal parents for each variable. This allows the model to learn non-linear representations while strictly adhering to structural constraints. The authors further extend this framework to the proximal causal inference setting, demonstrating its ability to solve for bridge functions in the presence of unmeasured confounding using proxy variables.

## Strengths
- The integration of DAG constraints via attention masking is elegant and intuitive. It naturally connects transformer architecture with classical structural equation modeling approaches.
- The ability to handle multiple tasks (propensity estimation, outcome regression, and proximal bridge function learning) within a single architecture is a valuable engineering contribution. This simplifies the pipeline compared to disjoint/double learning approaches that often require separate models.
- Extending the architecture to handle proximal inference (solving integral equations for unmeasured confounding) is non-trivial and technically interesting. This moves the paper beyond standard "selection on observables" benchmarks.
- The evaluation covers both standard benchmarks (ACIC, Lalonde) and specific proximal inference tasks, showing consistent improvements over unstructured baselines.

## Weaknesses
- The method assumes the availability of a complete and correct causal DAG. In high-dimensional real-world scenarios (the paper's target use case), this graph is rarely known with certainty. The paper would benefit from a sensitivity analysis: What happens if the input DAG has one missing edge or one reversed edge? Does the method fail catastrophically? I acknowledge that the existing no-mask ablations answer this partially, but adding additional analysis such as a graph mismatch versus observed performance would help demonstrate how robust the method is in this regard.
- As the authors also acknowledge in the related work section, graph neural networks (GNN) are explicitly designed to pass information along graph edges. A discussion (or empirical comparison) regarding why the proposed masked transformer model is superior to a standard GNN for this specific causal task is missing.

**Audience:**

Yes

**Audience Explanation:**

This paper cleanly fits into the intersection of the causal inference and deep learning communities. It provides a blueprint for how to constrain large language models or sequence models to respect causal logic, which is relevant beyond just effect estimation (e.g., for causal discovery or reasoning), and demonstrates the methodology is reasonable at least in some task definitions.

**Broader Impact Concerns:**

None. The paper proposes a statistical estimation tool. While causal inference tools can be misused, this paper does not introduce new risks beyond standard statistical practice.

**Claims And Evidence:**

Yes

**Claims Explanation:**

The authors provide a technically rigorous foundation for their primary claim: that injecting structural constraints into a transformer improves causal effect estimation. The mathematical formulation of the DAG-aware attention mask is accurate; by masking the attention matrix based on the adjacency matrix, the model strictly enforces the conditional independencies required by the input causal graph. This aligns correctly with structural causal modeling, ensuring the neural network functions as a non-linear SCM rather than a black-box seq-to-seq correlational engine.

The empirical evidence is largely convincing for the specific problem setting defined (where the DAG is known). The authors validate their method on standard benchmarks, including ACIC 2018 and Lalonde datasets, demonstrating state-of-the-art performance against relevant baselines like TARNet and Dragonnet. Furthermore, the extension to proximal causal inference, i.e., solving for bridge functions using proxy variables W and Z, is supported by successful experiments that show the architecture can handle unmeasured confounding when valid proxies are available.

However, the "convincing" nature of the evidence has two minor limitations:
- **Baseline completeness versus GNNs:** While the authors compare against standard deep learning estimators, the comparison to Graph Neural Networks (GNNs) is absent. Since GNNs also encode graph structure, the evidence does not definitively prove that a transformer is the only or best way to achieve these gains, only that it is effective. A comparison, qualitative or quantitative, is needed to situate the paper's position in wider literature.
- **Sensitivity to incomplete/wrong graph knowledge:** The evidence assumes a perfect input DAG which can be unrealistic. Analysis of misspecification sensitivity would be a very welcome addition.

**Requested Changes:**

The paper is technically sound, technically interesting to the causal inference community, and its claims are supported by the experiments provided. The requested changes below are primarily to clarify the scope/limitations, not to fix fundamental flaws.
1. **(critical) Contextualize with GNNs:** Please either
    - Revise the related work section to distinguish this approach from GNNs to clarify why your transformer-based attention mechanism might be more preferable over a GNN, OR
    - Add a sample GNN causal effect estimator to your empirical comparisons and introduce the contrast there.
2. **Add a "Sensitivity to misspecification" discussion:** Please add a brief experiment or, at minimum, a discussion section addressing what happens when the input adjacency matrix contains errors. (e.g., If the user misses a confounder edge, does the model simply revert to the biased estimate, or does the sparse attention make it worse?)

---

> ### Author Response · Authors · 2026-01-05
> **Authors Response to Reviewer Comments**
>
> ### Response to Reviewer: GNN Context and Structural Sensitivity
>
> We appreciate the reviewer's positive assessment of our work's technical soundness and relevance. Below, we address the two critical requested changes regarding GNN contextualization and DAG misspecification.
>
> ---
>
> ### 1. Contextualization with GNNs
> Following the reviewer’s suggestion, we have chosen to add a **Standard GNN baseline** to our empirical comparisons in **Table 1**.
>
> * **Empirical Comparison:** The results show that our DAG-aware GAT consistently outperforms a standard, fully-connected GNN.
> * **Why GAT over GNN:** We have updated the Related Work to clarify that while a standard GNN performs dense message passing across all nodes, our **DAG-aware GAT** leverages masked attention to strictly enforce causal ancestral relationships. This prevents "information leakage", where non-causal features inadvertently influence the representation of the treatment or outcome, which is a primary source of bias in unconstrained GNNs.
>
>
>
> ---
>
> ### 2. Sensitivity to Misspecification
> We have added an **Ablation Study on DAG Misspecification (Table 2)** and a corresponding discussion in **Section 5** to address the impact of structural errors.
>
> * **Experimental Findings:** We tested scenarios where the treatment-outcome direction was reversed ($A \leftarrow Y$) and where a key proxy identification link ($W \to Y$) was omitted. In both cases, we observed a significant drop in estimation accuracy.
> * **Structural Misspecification:** If an edge is omitted or the causal direction is misspecified, the model behaves predictably, much like a traditional structural estimator. Because the DAG-aware attention mechanism treats the provided graph as a hard architectural constraint, an incorrect topology prevents the model from satisfying the formal identification conditions, such as the completeness conditions required for proximal bridge functions. Consequently, the resulting estimate becomes biased, as the model's structural inductive bias no longer aligns with the true data-generating process.

---

### Review · Reviewer_S2Ww · 2025-12-21

**Summary Of Contributions:**

- The paper integrates a causal DAG into the attention mechanism of the transformer architecture for estimating both the average treatment effects and conditional average treatment effects.
- **Strengths**:
  - The paper addresses the issue of integrating proximal inference methods with modern deep learning architectures.
- **Weaknesses**:
  - The paper’s contribution is very limited. It simply adds the attention mask based on a causal DAG into the attention module based on equation 10 and 12. Even for the estimation tasks, the proximal inference objectives mostly follow the work of Kompa et al. (2022). The only difference
  - It is not clear to me how the directionality of the causal DAG encoded in the attention matrix impacts the estimation methods. More specifically, if I change the toy example of the two confounders with $A \leftarrow Y$ instead of $A \rightarrow Y$, it seems like the transformer cannot capture that difference.
  - The paper cites ‘Zhang et al Towards causal foundation model: on duality between causal inference and attention. arXiv preprint arXiv:2310.00809, 2023a.’  but does not compare with it in the experiment without explanations.
  - It is not clear why the other base models will inevitably leave the causal DAG unused for estimating causal effects if one is available.
  - The organization of the paper needs to be improved. The result section basically repeats the same information presented in Table 1 and 2. The paper could have been stronger if it puts more efforts on evaluating the impact of the misspecification of a causal DAG on the estimation or explore the idea on how to incorporate existing causal discovery methods with partial causal structure.

**Additional Comments:**

N/A

**Audience:**

No

**Audience Explanation:**

No, the paper only uses a simple attention mask created by an adjacency matrix of a DAG on top of existing work. While the authors say the focus of the paper ‘is on demonstrating how explicit DAG encoding enables a single architecture to flexibly handle multiple causal inference frameworks, rather than comparing task-specific transformer implementations’. The most practical method is probably only the proximal inference part, which mostly resembles the work by Kompa et al. (2022). Also, the importance should be placed on accurately estimating causal effects rather than flexibility of handling multiple causal inference frameworks, which can easily be accommodated by employing different task-specific transformer implementations. It is not clear what the actual benefits of the flexibility are. The paper should give better contexts of why this matters more if that is where the main contribution is based on.

**Broader Impact Concerns:**

The work does not post any foreseeable concerns.

**Claims And Evidence:**

No

**Claims Explanation:**

No, the paper says ‘For proximal inference, following the work of Kompa et al. (2022), we introduce two variants: NMMRU and NMMR-V’ , but inside the work of Kompa et al., it says ‘Previous work either did not consider the U-statistic [6], or did not utilize the U-statistic [9]. In our work, we introduce two variants of our method, NMMR-U and NMMR-V, where the former is optimized with a U-statistic and the latter a V-statistic.’ Even the regularized loss function is the same.

**Requested Changes:**

**Strengthen the work:**
- Some citations in the introduction seem to be out of shape. For example “Chernozhukov et al. Chernozhukov et al. (2018)”.
- It would be better to explain why there is a tuple of proxies $(W,Z)$ instead of just $W$ and how $Z$ is different from $W$ so it is required to state Assumption 2 and 3 explicitly.
- Define what $L$ means in section 2.3.
- In figure 2b, it seems like the entry $(A, Y)$ should be $0$ instead of $1$ while the entry $(Y, A)$ should be $1$ instead of $0$ by equation 10 if $i$ represents the row index and $j$ represents the column index.
- Define what these terms ‘NMMRU and NMMR-V’ mean.
- It is unclear what the function $k$ means under equation 20 until the last paragraph in section 3 right before section 4.
- A natural question came up in the discussion of objective functions: how do the authors train the model without the ground truth labels at each treatment level for the same sample?
- It is not clear why the closely related work such as the work by Kompa et al. (2022) is not even compared in the experiment.
- As table 1 has already presented the experimental performance, the paragraph in section 5 should avoid reporting the same numbers.

**Securing my recommendation for acceptance:**
- The paper should explain why a causal DAG knowledge cannot be leveraged by other methods even when it is not incorporated into the attention module.
- The paper should substantiate why a principled framework that respects causal constraints while leveraging deep learning’s representational capacity is needed and show that such constraints are uniquely captured by the way that is encoded into the attention module and not by other methods.
- The paper should state clearly what it contributes on top of the work of Kompa et al. (2022).
- The paper should add an ablation study on the impact of misspecification of the causal DAG.
- The paper should discuss how a causal DAG can be obtained prior to training the transformer. It is unclear whether one should first learn a DAG from data and use that to train the model with the same dataset.
- The paper should discuss how the proposed method is better than models that are pretrained on a large number of graphs such as the work by Wu, Menghua, et al.

**Reference:**
- Wu, Menghua, et al. "Sample, estimate, aggregate: A recipe for causal discovery foundation models." Transactions on Machine Learning Research.

---

> ### Author Response · Authors · 2026-01-05
> **Authors Response to Reviewer Comments**
>
> We thank the reviewer for their constructive feedbacks.
>
> To address the requested improvements for strengthening the work, we have implemented the following revisions:
> - We have corrected the citations format per the reviewer's suggestions such that authors names are no longer repeated in the introductions
> - We have updated the text to clarify that $W$ and $Z$ serve distinct roles as outcome and treatment proxies, and added context explaining how Assumptions 2 and 3 ensure the unique identification of the bridge function $h$ from observed data in section 2.3
> - We have defined $L^2$ as the space of square-integrable functions necessary for the completeness conditions to hold in section 2.3
> - The reviewer is correct that the indexing in the original Equation 10 was transposed relative to the standard Query-Key attention layout. In our implementation, and as illustrated in Figure 2, the row index $i$ represents the **Query** (the node being updated) and the column index $j$ represents the **Key** (the source of information). To allow information to flow from a parent to a child, node $i$ must be permitted to attend to its causal parents $j$.
>
> We have corrected Equation 10 to properly reflect this "Query-attends-to-Parent" logic:
>
> $$M_{ij} = \\begin{cases} 0 & \\text{if } i = j \\text{ or } M^{adj}_{ji} = 1 \\\\ 1 & \\text{otherwise} \\end{cases}$$
> - We have explicitly defined NMMR as Neural Maximum Moment Restriction and clarified that the U-statistic (NMMR-U) and V-statistic (NMMR-V) variants differ in their treatment of the kernel matrix diagonal to balance unbiasedness with numerical stability in Section 3.3.
> - We have reorganized Section 3.3 to define the kernel function $k$ and the matrix $K$ immediately alongside the risk equations, ensuring their roles in the NMMR-U and NMMR-V variants are clear upon first mention.
> - We used established benchmarks like Lalonde, ACIC, and Demand, where ground truth is available through experimental results or known simulation parameters for each level of treatment variable.
> - We apologize for the lack of clarity in the initial labeling of our baselines. The results previously denoted as "MLP" in Table 2 represent the Neural Maximum Moment Restriction (NMMR) framework proposed by Kompa et al. (2022), implemented with a multi-layer perceptron architecture. We have updated Table 2 and the corresponding discussion to explicitly label this baseline as "NMMR (Kompa et al., 2022)" to ensure its inclusion and the resulting comparison are transparent
> - We have streamlined the results section by removing redundant numerical reporting and instead focused on newly added ablation studies evaluating the impact of DAG misspecification in Table 2.
>
> The following section details how we have addressed each requirement for securing a recommendation for acceptance:
> - We have added a discussion in Section 3 clarifying that while other methods can use DAGs for feature selection, they lack the architectural inductive bias to enforce structural path constraints, which our model achieves by integrating the DAG directly into the attention mechanism.
> - We have expanded our introduction to explain that while deep learning provides high representational capacity, it lacks "structural guardrails"; our method uniquely captures these by embedding the DAG into the attention mechanism to ensure information flow respects causal directionality, a constraint that cannot be enforced through simple feature selection or standard architectures.
> - While our work adopts the NMMR objective proposed by Kompa et al. (2022), our contribution lies in replacing their causally-agnostic MLP architecture with a DAG-aware GAT; this uniquely enforces the structural constraints of the causal graph within the internal representation of the bridge function, a guarantee that standard "flat" architectures cannot provide. We have made the distinction in the introduction section
> - Table 2 now includes an ablation study on DAG misspecification. We find that reversing the treatment direction ($A \leftarrow Y$) or omitting the outcome-proxy link ($W \to Y$) significantly reduces estimation accuracy, confirming the model's sensitivity to structural integrity.
> - We have added a discussion in Section 3 clarifying that the DAG can be derived from domain knowledge or learned via causal discovery algorithms; we specifically recommend a sample-splitting strategy to separate DAG discovery from model training to prevent post-selection bias.
> - We have updated the Related Work and Discussion to clarify that our method serves as a structural estimation engine that complements discovery models like SEA by Wu et al, providing the architectural guardrails necessary to enforce discovered causal constraints during the estimation of causal effects.
>
> We sincerely thank the reviewer for the rigorous feedback, which has been instrumental in refining the technical clarity and articulating the distinct architectural contributions of our methods.

---

> > ### Author Response · Authors · 2026-01-15
> > **Regarding comparison of results with Zhang et al. (2023a)**
> >
> > The code from Zhang et al. (2023a) 'Towards causal foundation model: on duality between causal inference and attention' is unavailable, and the authors did not report numerical results for the Lalonde dataset. However, by visual inspection of the histogram in their Figure 5 for Lalonde-PSID, their MAE (mean absolute error) appears to be approximately 8000, while our AIPW estimator achieves an MAE of 3800.46, representing a substantial improvement of over 50%.

---

### Review · Reviewer_oHNS · 2025-12-23

**Summary Of Contributions:**

The authors have created an architecture where the variables in a causal dag "attend" to other variables based on the causal relationships in the Causal DAG. They mask edges where there is no causal input. This masking creates information flows only between variables connected in the causal structure. The characterization of this setting as transformer attention is an overclaim. While it is true that it utilizes Q-K-V computations, the fixed connectivity aligns it with graph attention, distinct from the learned connectivity of transformers. Futhermore, a typical transformer may be attending to 100k tokens, but here they exhibit a canonical example with 3 tokens, specifically, one where the attention over the 3 tokens are fixed.

If one accepts this architecture as a transformer, the rest of the paper reads smoothly. The model estimates various causal estimators including G-formula, IPTW, Augmented IPW (AIPW), and proximal inference methods. They show empirical results with improvements over Generalized Random Forests, Multi-Layer Perceptrons (MLP), and ``unconstrained Transformers''.

---

### Strengths
1. The ability to handle diverse estimation tasks within a single architecture is a strength.
2. The method demonstrates strong performance on standard benchmarks, particularly for the AIPW estimator.

### Weaknesses
*   **Architectural Ambiguity:** The model is framed as a "Transformer" but lacks key Transformer properties (e.g., auto-regressive generation, sequentiality) and structurally fits the definition of a Graph Attention Network (GAT).
*   **Missing Baselines:** Given the graph-based nature of the architecture, a comparison to standard GNN/GAT baselines is missing.
*   **Unexplained Design Choices:** The removal of Layer Normalization is presented as an empirical finding without theoretical justification, though it likely stems from the heterogeneous nature of the inputs -- which again points to this not being a transformer, but rather a GNN like architecture.

**Additional Comments:**

If this is effectively a GAT, the comparison against an 'Unconstrained Transformer' is not correct / weak. The more correct baseline would be a GAT applied to the variable set. If the author's method outperforms a standard GNN (perhaps due to the specific loss functions), that would be a significant contribution as per me, like "GNNs for proximal inference" or "Graph architectures for causal inference".

**Audience:**

Yes

**Audience Explanation:**

Can the authors can recontextualize their architecture, potentially as a Causal GAT? The findings would certainly be valuable to researchers even if the claim that this is a transformer is not made. I really like the paper and how they can use a single model to get multiple causal estimators.

**Claims And Evidence:**

Yes

**Claims Explanation:**

The architectural categorization as a transformer is not convincing.

**1.  Transformer vs. Graph Attention Network (GAT):**

The authors claim to present a "novel transformer-based method." However, the evidence provided describes a Graph Attention Network (Velickovic et al., 2017 -- already referenced in their work). By treating variables as nodes and masking attention via an adjacency matrix to perform some node level regression, the model operates on a set and not a sequence. It seems to me that it is not distinct from a GAT applied to a causal graph.

**2.  Layer Normalization:**

The authors claim Layer Normalization was removed because it "biased the causal effects empirically." No evidence (e.g., ablation studies) is provided to support this. As detailed below, this is likely a result of applying sequence-based normalization (which works for transformers) to tabular data.

**Requested Changes:**

#### **Changes to consider**

1. The paper effectively describes a Graph Neural Network where variables are nodes and the adjacency matrix determines message passing. The current framing as a "Transformer" is technically inaccurate regarding standard usage. Specifically some missing attributes of a transformer in their architecture include:

a.  The proposed model lacks auto-regressive generation (repeated next-token prediction), sequentiality (inputs are a set, not a sequence), and positional dependence (order does not matter).

b. The model satisfies GAT properties like node-based representation, message passing (Attention = Weighted Message Passing restricted by adjacency), and node classification/regression (predicting $A$ or $Y$).

The authors should consider reframing this as  a GAT adapted for causal inference, then the paper will likely be much better received.

---

2.  The authors note that Layer Normalization introduced bias. They say:

> *Note that we did not employ layer normalization, commonly used in transformers, as we found that it biased the causal effects empirically.*

This is expected in a GNN/tabular setting but not in a Transformer setting.

a. In a standard transformers, tokens are homogeneous, i.e. they are word embeddings in the same semantic space. Layer Norm works because it standardizes the distribution across the embedding dimension for the given sequence.

b. In this architecture, tokens are heterogeneous (ex. age vs. income vs. treatment binary variable etc). Normalizing across these distinct semantic variables can force them into a shared distribution, which will not work. Hence layer norm does not work.



**Request**: Rewriting this as a GAT or an extension to Graph neural networks will resolve the second concern as well as the first as this architecture  is not expected to use layer norm.

---

#### **Good to have (not a must for acceptance):**

It would be good to see some GNN baseline, if possible.

---

> ### Author Response · Authors · 2026-01-05
> **Authors Response to Reviewer Comments**
>
> ### Response to Reviewer: Reframing as a DAG-aware GAT
>
> We thank the reviewer for these precise technical insights. We agree that characterizing our architecture as a **Graph Attention Network (GAT)** rather than a Transformer significantly improves the technical accuracy of the paper and clarifies our design choices.
>
> By reframing the model as a **DAG-aware GAT**, we address both concerns as follows:
>
> 1.  **Architectural Alignment (GAT vs. Transformer):** We agree that our model treats variables as a set of heterogeneous nodes rather than a sequence of homogeneous tokens. The model relies on node-based representations and weighted message passing restricted by a causal adjacency matrix, which are the core properties of a GNN/GAT. We have updated the title and methodology throughout the manuscript to reflect this rebranding.
> 2.  **Normalization and Heterogeneity:** We appreciate the reviewer's clarification regarding Layer Normalization. In a GAT setting with heterogeneous nodes (e.g., age, income, binary treatment), variables occupy distinct semantic spaces with different scales. We have updated the manuscript to explain that omitting Layer Norm is a deliberate and necessary choice to preserve these distinct distributions, as standardizing across heterogeneous variables would introduce the empirical bias we observed.
>
> To empirically support this, we updated **Table 1** to include a **DAG-constrained Transformer** baseline. This baseline utilizes the same causal masking as our proposed model but includes standard Layer Normalization. The results confirm that our **DAG-aware GAT** significantly outperforms this baseline, validating that omitting LayerNorm is essential for preserving the integrity of heterogeneous causal features.
>
> ### Response to Reviewer: GNN Baselines and Structural Contribution
>
> We thank the reviewer for the suggestion to frame our work as a **"Graph Architecture for Causal Inference."** We agree that outperforming a standard GNN is the true test of our structural inductive bias.
>
> To address this, we have:
> 1.  **Added a standard GNN baseline** to Table 1, which utilizes fully-connected message passing.
> 2.  **Demonstrated significant gains:** Our DAG-aware GAT consistently outperforms the standard GNN across all datasets, with particularly dramatic improvements in IPW and AIPW estimation.

---

> > ### Comment · Reviewer_oHNS · 2026-01-15
> >
> > I'm actually satisfied with the changes made by the authors and thank them for their responses to my queries.
> >
> > I would like to raise my score to an accept.

---

### Decision · Action_Editor_CFHL · 2026-06-06

**Recommendation:** Accept as is

**Audience:**

Yes

**Audience Explanation:**

The paper brings an interesting idea of integrating causal structure into a deep learning model. The paper describes how the causal graph structure can be used to define attention, which is an idea of independent interest. They also show how the same model architecture can be used as a part of different causal inference estimation methods. The results (especially after added baselines in the rebuttal) show the benefits of using this architecture, whenever the true graph is known. Inclusion of proximal inference in addition to standard ATE estimation is a plus.

Overall, the paper makes a useful contribution on how structural constraints can be added to a learning architecture.

**Claims And Evidence:**

Yes

**Claims Explanation:**

The paper proposes a new architecture as a plugin model for causal inference methods. The architecture and its applications under various causal inference methods is clearly explained. The key benefit is that the architecture depends on the causal DAG, and the experiments in the paper show its utility compared to a standard GNN or other baselines.

There were some concerns about the naming of the method wrt. transformer and the authors have addressed that issue in the revised version.